# "We must work tirelessly to promote …": Mediating interpersonal commitment to material processes in English translation of Chinese political discourse

Jiachun Li[1], Shukun Chen[2]*, Yawen Zhang[3]

**1** School of Western Studies, Heilongjiang University, Harbin, Heilongjiang, China, **2** School of Foreign Languages and Cultures, Guangdong University of Finance, Guangzhou, Guangdong, China, **3** School of Interpreting and Translation Studies, Guangdong University of Foreign Studies, Guangzhou, Guangdong, China

* shukunchen@gduf.edu.cn

## Abstract

Evaluative items have been central in political translation studies. This paper extends previous research by focusing on "circumstances + material processes" expressions through data extracted from Chinese-English parallel corpus of *Xi Jinping*'s discourse. Using re-instantiation as its descriptive framework, this research imported into Excel 1686 concordance lines for annotation to reveal how and why certain translation shifts occur. Results show that a clear shift occurs in the translation of the circumstances. Zero forms are employed to reduce interpersonal commitment in translation, while verbs and adjectives are more likely to be used to raise it. The study identifies rhetorical and political motivations for these translation choices. It is hoped that the linguistics-grounded corpus approach provided in this paper could be applied to translation research on other grammatical domains or text types.

## Introduction

Political texts rich in ideological and cultural nuances often pose a unique challenge to translation practitioners who "exercise the power of modifying the text/narrative in hand in accordance with their perceived ideology" [1: xiv] Mediation of power and ideology in translation could be identified through textual analysis with linguistic tools such as critical discourse analysis and systemic functional linguistics (hereafter SFL) [2]. The notion of interpersonal meaning/function provided by SFL has been particularly useful in this line of research. Interpersonal meaning refers to the use of language to establish, maintain, and negotiate social relationships and interactions between individuals. This involves expressing attitudes, opinions, emotions, and intentions while considering the context, power dynamics, and cultural norms within

**Data availability statement:** All relevant data are within the manuscript and its Supporting Information files.

**Funding:** The study is funded by "Project of Guangdong Provincial Philosophy and Social Science Planning 2024: A Study on the Projection Language Features in Political Discourse and Their Translation [Project Number: GD24CWY13]". The funder had no role in study design, data collection and analysis, decision to publish, or preparation of the manuscript.

**Competing interests:** The authors have declared that no competing interests exist.

a given communicative situation [3,4]. This perspective has effectively informed descriptive translation studies in the political context.

Some scholars investigate distributive patterns of interpersonal components such as modality [5–7] and pronouns [8] in political translation. These studies reveal both linguistic features of Chinese political discourse as well as the translator/interpreter's political awareness. Fu [5] analyzes the data from government press conference interpreting and finds that the interpreter tends to push up the degree of modal verbs, for instance, from "*yao* 要 (need)" to "must". Yu & Wu [8] examine the pronoun "we" in the translation of government reports, showing how the pronouns are added in translation to project the government image as a transforming agent.

Another strand of scholarship employs the notion of grammatical metaphor in SFL to discuss value-laden items [9,10]. Zhang [9] analyzes how implicitness is conveyed in diplomatic interpreting, arguing that some information is concealed through this linguistic mechanism for national interests. A case in point is that when "it is impossible" is used to render the meaning of "could not", the attitude of the spokesman could be covered up, leaving multiple readings for an audience.

The appraisal theory, a framework within SFL that is concerned with how language is used to assign value to experiences, events, and states of affairs [4], has guided a number of political translation studies [11–14]. Lexical choices have been analyzed to tackle issues such as stance-taking or national image construction. Li and Pan [14] identify adjectives and adverbs related to China, finding that China is more negatively constructed in translated versions, probably to "keep a low profile".

Items such as adjectives and adverbs are of particular importance for translation studies in political contexts because, on the one hand, they are evaluative resources that pertain to attitude and ideology, and, on the other hand, they rhetorically tend to be overused, which should arguably be eliminated in translation for better communication effects [15,16]. Therefore, this study is part of a long-term project investigating how attitudinal adverbs (or circumstances) are coupled with various types of verbs (or processes) in Chinese political discourse and how they are translated in institutional practice. Our previous research already observes the coupling of "manner + mental process", finding that translation choices are highly sensitive to the occurrence of adverbs, and evaluative forces of those adverbs are retained in translation through specific linguistic mechanisms [17]. However, whether such a conclusion could be generalized to other types of processes remains unknown. This study will focus on material processes and expand its scope of observation to include circumstances, namely, adverbs and prepositional phrases that represent the meaning of force.

The "material process" was relatively neglected in previous quantitative studies. In SFL, it refers to a type of clause that concerns actions, happenings, and events that involve creation or transformation in the external world, typically expressed by verbs such as "build", "promote", "develop", or "complete", as shown in example (1) below ("[[…]]" is used to mark the process in the source text). Material processes are vital in political discourses as they are highly motivating and help construct the government as a transforming agent. (Yu & Wang 2023)

(1) 全面　　[[推进]]　　依法治国
　　fully　　promote　　law-based governance of the country

We analyze data extracted from the Chinese-English paralleled corpus of *Xi Jinping: The Governance of China* to answer the following research questions:

(1) What are the translation choices for circumstances, in terms of grammatical forms and interpersonal commitment?

(2) To what extent are the translation choices sensitive to the occurrence of circumstances coupled with material processes in Chinese political texts?

(3) How are the translation choices motivated in light of the political contexts?

This study is an attempt to contribute to the interpersonal line of research on political translation by (i) adopting the theoretical model of translation in SFL, namely, treating translation as re-instantiation, to prove its explanatory applicability, (ii) demonstrating the linguistic features of Chinese political discourse from the perspective of process types, and (iii) offering a linguistic insight into the translation choices of arguably redundant interpersonal items.

## Literature review

There has been extensive literature on political translation during the past several decades [2,7,10,18–20]. Overall, two strands of research can be identified: (i) the role of translation in political communication through observing significant historical/cultural events [19], and (ii) translation choices concerning political discourse features [2]. This study is situated in the second line of research, which could be explored with two approaches, namely, prescriptive and descriptive [21].

The prescriptive studies underscore what are the proper translation strategies or principles in the political contexts. Some scholars studied the methods and techniques for translating political terminologies such as hot words [10], metaphors [9,20,22], and nominalization [23]. For example, Zhu and Zhang [10] analyze the translation strategies of "*buzheteng* 不折腾 (no trouble-making)" and advocate using zero translation to achieve the effects of defamiliarization. Pan et al. [20] examine the mediation of political metaphors in media discourse.

Others [16,18,24] aim to inform the practitioners of what they should do in political translation in light of rhetorical features and communication effects. Chen [24] points out that the ultimate aim of translating the political texts of the CPC is to achieve good effects in international communication. Two important points should be borne in translators' minds: one is to translate accurately and faithfully, and the other is to translate in an easily accessible way. Numerous studies have shown that faithfulness and acceptability are important criteria in political translation. Some scholars probe into classic political literature such as *Selected Works of Mao Zedong* [25] and evaluate the quality of translation.

Descriptive studies offer detailed description of what really happened in political translation, mostly relying on a corpus method. The data generated out of a large-scale corpus provide insights into linguistic features of various political texts and patterns of translation choices [26–28]. Translation studies of interpersonal resources have been particularly fruitful in this respect [7,8,11–14]. Yu and Wang [7] find that in Chinese-to-English translation, high-valued modal expressions in the source texts are often omitted or translated into lower-valued expressions, while median-valued modal expressions tend to be retained. Additionally, the translated texts exhibit a shift towards a more target-oriented tone aligned with English communicative conventions and the intended persuasive function in the target culture. This study adopts exactly this descriptive approach to examine interpersonal shifts in political translation.

It should be noted that in recent years, an increasing number of scholars analyzed data from the *Xi Jinping: The Governance of China* corpus, with particular focus on evaluative items such as political metaphors [29] and adverbs [17], shedding new light on the institutional translation practice. Their major findings are that evaluative forces tend to be mediated in translation for reasons such as image shaping, acceptability, and nuanced equivalence.

Some limitations can be identified in the previous research. First, previous studies have not drawn heavily upon a meaning-oriented linguistic theory to guide their analyses. A linguistic model of translation could well supplement previous research by offering a more accurate description and a replicable methodology. Second, extant corpus studies have covered evaluative items such as pronouns, modal verbs, and adjectives. However, certain couplings or constructions such as "adverbs + verbs" in Chinese political texts have not drawn enough attention. Therefore, this study aims to extend previous research by providing a more linguistically grounded analysis of translation shifts in "circumstances + material processes" under the theoretical framework of re-instantiation based on data from an exemplar corpus.

## Theoretical framework

This study will describe the translation choices using the theoretical framework of re-instantiation, i.e., treating translation as two instances of texts that share meanings to a certain degree. In SFL, instantiation refers to the process whereby a general linguistic system is realized in specific instances of language use. The theory posits that language is a resource for making meaning, and it does so through a system of choices at various levels, including lexico-grammar (vocabulary and grammar) and discourse semantics (how meanings are created across clauses and sentences). SFL recognizes a hierarchy of instantiation, ranging from the most general to the most specific. Scholars emphasize the usefulness of this concept in describing the relationship between texts such as paraphrasing, retelling, etc [30]. More specifically, the instantiation cline involves system, genre/register, text type, text, and reading, where system represents the most general end of meaning potential and reading stands for subjectified meaning, the most actual end. (See Fig 1)

Martin [31–33] coins two terms to describe intertextual relations, viz, coupling and commitment. Coupling is used to refer to "the ways in which meanings combine, as pairs, triplets, quadruplets or any number of coordinated choices from system networks" (p. 491). In example (1), "*quanmian tuijin*全面推进 (fully promote)" can be considered as a case of coupling between evaluative meaning of force, a choice from the system of graduation, and experiential meaning of doing, a choice from the system of transitivity. Commitment refers to the amount of meaning potential activated in a particular process of instantiation. Consider example (1) again. "*Quanmian tuijin*全面推进 (fully promote)" is more specific than "*tuijin*推进 (promote)" since the former involves more interpersonal commitment with "*quanmian 全面* (fully)" to evaluate the scope of the action "promote".

It is useful to model translation as a process of re-instantiation because translation is a meaning-creating process guided by the source text [34,35]. Numerous scholars have attempted to employ the framework of re-instantiation for translation research [17,36–38]. Coupling and commitment are used to analyze micro-strategies for equivalence or shifts in translation. Chang [18] observes the data from Jane Austen's novel *Pride and Prejudice* and its translated versions, focusing on different degrees of ideational and interpersonal commitment. He finds that various translation and adaptation techniques such as changes in the grammatical metaphor and projection would bring different reading experiences to the readers. Particular value should be assigned to the source text in literary texts. The notion of coupling is useful in

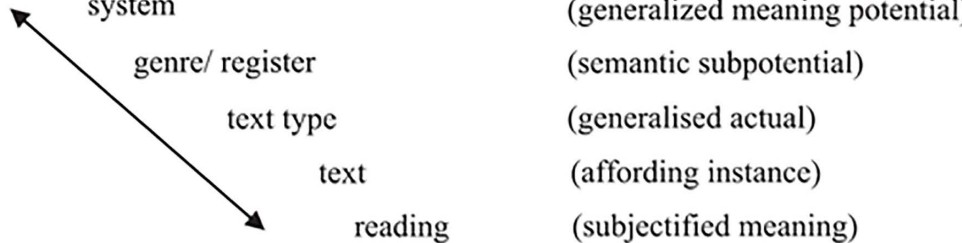

**Fig 1. Instantiation as a cline from reading to system (Martin [30]).**

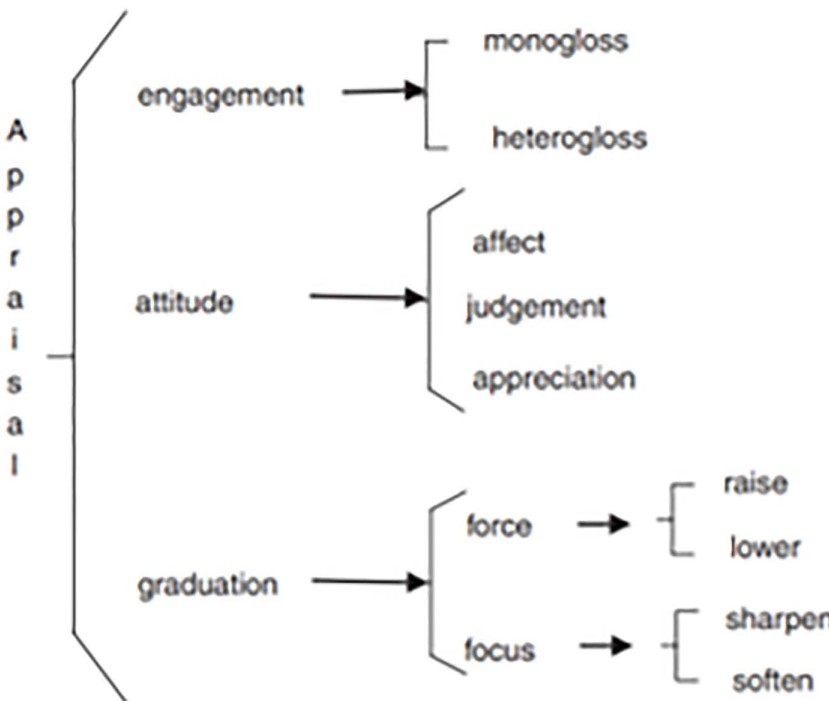

describing how interpersonal meaning and choices of ideational meaning are combined in source texts compared with the translated [17]. For this study, it is necessary to briefly introduce the system of transitivity and graduation.

In SFL, along the experiential line of meaning, the system of transitivity recognizes different "process types", which are used to refer to the way clauses are structured to represent different kinds of processes in the world [3]. These processes are the building blocks of grammar and reflect our experience of events, states of being, and relations between entities. There are six process types commonly recognized in SFL: material process, mental process, verbal process, relational process, behavioral process, and existential process. The process in focus is a material process, which represents actions or happenings that involve physical movement or change. They often involve an Actor (the doer of the action) and a Goal (the entity affected by the action).

Transitivity system also involves various types of "circumstances", the components of a clause that provide information about the conditions or environment in which a process takes place. These are traditionally referred to as adverbials and can be realized through various forms such as prepositional phrases, adverbs, or finite clauses. The types of circumstances relevant to this study include frequency, extend and manner in the "enhancing circumstances" [3].

We also refer to the Appraisal Theory [39] to examine the interpersonal meaning in translation. In particular, we considered the system of graduation and attitude, as displayed in Fig 2. Graduation is one of the three main systems of meanings used to express positive or negative evaluations in discourse. Graduation deals with adjusting the strength or intensity of an evaluative stance. It operates along two axes: force and focus.

Force concerns the intensity or amount of the evaluated phenomenon. Force can be either "raised" (upscaled) or "lowered" (downscaled). For example, the words "very," "extremely," or "somewhat" can be used to modify the force of an adjective or adverb.

**Fig 2. An overview of appraisal resources from Martin & White (2005: 38).**

Focus sharpens or blurs the boundaries of the evaluative term. Under focus, we can distinguish between "sharpening" (specifying a term more precisely) and "softening" (vagueness or imprecision). For instance, words like "real" or "true" can sharpen the focus of a category, while "sort of" or "kind of" can soften it.

The Attitude subsystem represents choices of expressing feelings and evaluations. It is divided into three core domains: Affect, which deals with emotional responses and feelings; Judgement, which concerns the ethical assessment of behavior and character; and Appreciation, which evaluates the aesthetic and social value of objects, texts, and phenomena.

In the case of the current study, we aim to observe how the meaning of force and attitude has come to enhance the material processes in Chinese political texts, and the correspondent translation choices. That means, only those circumstances that carry the meaning of evaluative force (i.e., degree, frequency and manner) are included for the current study.

## Methodology

We chose the corpus of *Xi Jinping: The Governance of China* (http://imate.cascorpus.com) as the source of materials. The corpus was based on the book series *Xi Jinping: The Governance of China* (I-IV), which consists of Chinese President Xi Jinping's major publications, including speeches, correspondence, articles and interviews. The works were translated to English by Chinese and foreign translators and experts organized by the Chinese government. This ensures the translations represent institutional practices of the highest standard in Chinese political discourse. Therefore the texts can guarantee that our findings can be generalized to the whole political genre. The corpus has a storage capacity of 359,546 characters in Chinese and 459,343 in English words. Despite its strengths, the corpus has limitations. First, the corpus is limited to Xi Jinping's discourse, excluding other political figures or texts, which may restrict the generalizability of findings to broader Chinese political discourse. Second, manual annotation of grammatical forms and interpersonal commitment risks interpretive bias.

We downloaded the wordlist and selected ten verbs that represented state change events based on frequency of occurrence in the corpus and their corresponding translated versions.

These ten verbs were chosen for the following reasons:

1) In Chinese, the event of caused state change is frequently encoded by "verb + complement" constructions, e.g., "*tuijin* 推进 (push + enter)", and these ten verbs are common in daily usage;

2) The ten verbs express different meanings, but they are all in the same semantic domain of caused state change;

3) The frequencies of these verbs were high enough to generate data that can reveal patterns of translation shifts;

4) Some of the processes expressed by these verbs are coupled with circumstances and some are not, so it is interesting to see the interaction between circumstances and processes as well as the differences in translation.

We further screened the data according to the following criteria: 1) the verbs should be process (the main predicate) of the clause, not a component in the nominal phrase or adjective phrase; 2) they should be a single process of the clause (because they might be omitted in the English version if they occur with other verbs and express similar meanings); 3) they should encode the component of cause, thus expressing events of caused state change. The ten key words generated 1686 concordance lines (Table 1 shows the respective numbers of the concordance lines).

The results of searching were imported to Excel for annotations of translation choices (see Fig 3). We manually coded the data in terms of grammatical forms in source texts (ST) and target texts (TT) and semantic change of interpersonal commitment. More specifically, we annotated each of the concordances for grammatical forms used in the translation of material processes, grammatical forms of circumstances in the ST, grammatical forms used in the translation of the circumstances and the status of interpersonal commitment (see Table 2 for annotating criteria and examples). It should be

**Table 1. Top ten verbs of material processes.**

| Material process | Frequency | Number of concordances | Material process | Frequency | Number of concordances |
|---|---|---|---|---|---|
| *tuijin* 推进 promote (push enter) | 500 | 287 | *jiancheng* 建成 establish (build become) | 197 | 100 |
| *tigao* 提高 improve (improve high) | 462 | 169 | *gaishan* 改善 improve (change good) | 127 | 270 |
| *cujin* 促进 promote (urge enter) | 398 | 61 | *kuoda* 扩大 widen (widen big) | 120 | 78 |
| *jiakuai* 加快 accelerate (add quick) | 350 | 31 | *jiada* 加大 broaden (add big) | 109 | 331 |
| *shenhua* 深化 deepen (deep-melt) | 275 | 82 | *zhuahao* 抓好 grasp (grasp good) | 99 | 277 |

noticed that the annotation of "zero form" means the original component was completely omitted in translation. The annotation "n/a" means the component was not found in the ST, and hence not found in the TT. Therefore, "zero form" should be considered as a translation choice while "n/a" should not.

For objectivity of the results, the authors of this paper annotated all the concordance lines separately and resolved all the discrepancies. The intercoder discrepancy rate was 1.06%, indicating high agreement. The next step is to calculate the frequency and percentage of the annotations via Excel for subsequent analysis.

## Quantitative findings

In order to find out whether and to what extent the translation choices are sensitive to the occurrence of circumstances, we first divided the processes into material processes without circumstances and those with circumstances. Then, we calculated the translation choices of the two conditions, to see the influence of circumstances on the translation of processes (see Table 3).

From Table 3, we can see the distribution of different translation choices as reflected in grammatical forms. The frequency and percentage of the grammatical forms show similar patterns in translation of material processes both with and without circumstances, indicating that translation choices of processes are not sensitive to the occurrence of the circumstances, which is contrary to the findings of our previous research on the occurrence of manners with mental processes [17]. This might suggest that, in translators' considerations, the coupling between circumstances and material processes (e.g., "actively pursue") is more readily acceptable for the target audience than that between manners and mental processes (e.g., "deeply understand").

Provided that the occurrence of circumstances would not affect the translation choices of material processes in general, we concentrated on the translation choices of the 413 instances of circumstances. The circumstances in our data all carry the evaluative force in the sense of frequency, degree and manner. The top five circumstances are *buduan* 不断 (continually), *quanmian* 全面 (fully), *jiji* 积极 (actively), *xietiao* 协调 (coordinately), and *qieshi* 切实 (practically), representing 28.33%, 11.62%, 4.35%, 3.87%, 2.91% of the total respectively. These circumstances boost the force of the material processes, thus raising the mobilizing effects of the political communication.

The coupling between material processes and various circumstances seems to be a prominent feature of Chinese political texts as observed by several scholars [18,25]. Translation shifts are frequently found, as shown in Table 4.

**Fig 3. A snapshot of annotations for translation choices via Excel.**

Table 4 displays the distributions of the grammatical forms that realize circumstances in ST in relation to the forms used in translation. In the source text, the major form of circumstances is adverbs (e.g., *buduan* 不断 continually), representing 81.11%, followed by prepositional phrases (e.g., *yi zhizhengzhaoxi de jingshen*以只争朝夕的精神 with spirit of seizing the day), 8.47%, idiomatic phrases (e.g., *jianding buyi*坚定不移 resolutely without moving), 6.54%, and complexes (e.g., *jiji wentuo*积极稳妥 actively and steadily), 3.87%.

In translation, eight grammatical forms were used to re-instantiate the circumstances, namely, adverbs, verbs, prepositional phrases, adjectives, nouns, complexes, idiomatic phrases, and zero form. Translation shifts can be clearly observed in terms of grammatical forms. The choice of zero form accounts for 42% of the total, meaning that omission was the major translation technique in translation of the circumstances. The second most frequent choice was adverbs (19%), followed by verbs and prepositional phrases (both taking up 16%). The choices of adjectives, nouns, complexes and idiomatic phrases represent less than 5%. This indicates that translators chose various grammatical forms to re-instantiate evaluation, thus mediating the force of material processes in translation.

The translation choices of the 413 instances of circumstances in terms of interpersonal commitment are shown in Table 5. It shows the top five frequent grammatical forms in translation in relation to equal, upscaling and downscaling of interpersonal commitment. The major patterns are found as follows.

(i)  Zero form represents downscaling of interpersonal commitment.

(ii)  Other than zero form, equal commitment is the dominant choice of translation.

(iii) Verbs and adjectives tend to be used to raise interpersonal commitment, with 28% and 31% of likelihood, respectively.

**Table 2. Annotating criteria for the translation choices of material processes and circumstances.**

| Observed phenomena | Annotating Criteria | Annotation | Examples |
|---|---|---|---|
| translation of the material processes | the unit in English translation that semantically corresponds to the material processes in Chinese | adverbs | *shenhua* 深化 (deepen) → further (understand …) |
| | | verbs | *tuijin* 推进 (promote) → push forward |
| | | adjectives | *shenhua* 深化 (deepen) → deeper |
| | | nouns | *gaishan* 改善 (improve) → improvement |
| | | prepositional phrases | *jiancheng* 建成 (complete) → in place |
| | | zero form | *gaishan* 改善 (improve) → omission |
| circumstances in the ST | the circumstances coupled with material processes in Chinese; They should carry evaluative meanings in terms of force | adverbs | *buduan* 不断 (continuously)<br>*jiji* 积极 (actively) |
| | | prepositional phrases | *tongguo* 通过 (by)<br>*anzhao* 按照 (according to) |
| | | idiomatic expressions | *jianchi buxie* 坚持不懈 (persistently and not giving up)<br>*bushi shiji* 不失时机 (losing no time) |
| | | complexes | *yibubu zhashi* 一步步扎实 (step by step, solidly) |
| | | n/a | No circumstance was found in Chinese. |
| translation of the circumstances | the unit in English translation that semantically corresponds to the circumstances in Chinese | adverbs | actively, constantly |
| | | prepositional phrases | in a coordinated way, in accordance with |
| | | idiomatic expressions | step by step |
| | | complexes | actively and steadily |
| | | adjectives | coordinated, effective |
| | | verbs | work, make every effort |
| | | nouns | initiative, efforts |
| | | n/a | No circumstance was found in Chinese. |
| | | zero form | The circumstance was not translated. |
| interpersonal commitment (semantic change in translation) | same amount of evaluative meaning is activated in translation regardless of grammatical forms | equal | "*jiji* 积极 (actively)" → "actively"<br>"*xietiao* 协调 (coordinately)" → "in a coordinated way" |
| | more amount of evaluative meaning is activated in translation in terms of specificity, e.g., adding more evaluative items or being more explicit or direct in evaluation | up | "*buduan* 不断 (continuously)" → "work tirelessly to"<br>"*shenru* 深入 (deeply)" → "work hard to" |
| | less amount of evaluative meaning is activated in translation, e.g., omission or reducing evaluative items | down | "*buduan* 不断 (continuously)" → zero form<br>"*changqi jiji* 长期积极 (long and actively)" → "long" |

## Using zero form to downscale interpersonal commitment

Zero form for the translation of circumstances is self-evidently connected to the downscaling of interpersonal commitment as the meaning of intensification was lost in translation. In example (2), the material process coupled with the circumstance *buduan* 不断 (continually) in ST was translated into a single verb "promote" in TT, a typical case of omission in translation. Thus the process in the TT was downgraded in force.

**Table 3. The influence of circumstances on the translation of processes.**

| Processes / Translation Forms in translation of processes | Material processes without circumstances | | Material processes with circumstances | |
|---|---|---|---|---|
| | Frequency | Percentage | Frequency | Percentage |
| verbs | 1035 | 81.30% | 342 | 83% |
| adjectives | 25 | 1.96% | 8 | 2% |
| adverbs | 16 | 1.26% | 1 | 0% |
| nouns | 10 | 0.79% | 4 | 1% |
| prepositional phrases | 7 | 0.55% | 8 | 2% |
| zero form | 180 | 14.14% | 50 | 12% |
| Total | 1273 | 100% | 413 | 100% |

**Table 4. Grammatical forms of circumstances in ST vis-à-vis translation.**

| Grammatical forms | Circumstances in ST | | Translation of the circumstances | |
|---|---|---|---|---|
| | Frequency | Percentage | Frequency | Percentage |
| adverb | **335** | **81.11%** | 79 | 19% |
| verb | \ | \ | 67 | 16% |
| prepositional phrase | 35 | 8.47% | 66 | 16% |
| adjective | \ | \ | 16 | 4% |
| noun | \ | \ | 6 | 1% |
| complex | 16 | 3.87% | 4 | 1% |
| idiomatic phrase | 27 | 6.54% | 2 | 0.005% |
| zero form | \ | \ | **173** | **42%** |
| Total | 413 | 100% | 413 | 100% |

**Table 5. Grammatical forms in the translation of circumstances in relation to interpersonal commitment.**

| Interpersonal commitment | Grammatical forms in the translation of circumstances | | | | | | | | | |
|---|---|---|---|---|---|---|---|---|---|---|
| | n/a | | Adverb | | Verb | | Prep. phrase | | Adjective | |
| | freq. | per. | freq. | per. | freq. | per. | freq. | per. | freq. | per. |
| Equal | 0 | 0% | **71** | **90%** | 47 | 70% | **63** | **95%** | 9 | 56% |
| Upscaling | 0 | 0% | 0 | 0% | **19** | **28%** | 0 | 0% | **5** | **31%** |
| Downscaling | 173 | 100% | 8 | 10% | 1 | 1% | 3 | 5% | 2 | 13% |
| Total | 173 | 100% | 79 | 100% | 67 | 100% | 66 | 100% | 16 | 100% |

(2)

[ST]

就要　　不断　　[[推进]]　　实践　　基础　　上的　理论　　创新.

must　continually　promote　practice basis　on　theoretical innovation

[TT]

we should promote theoretical innovation based on practice.

## Using adverbs to achieve equal interpersonal commitment

Adverbs represent the second most frequent choice (19%) in translation. In different cases, the meanings of the circumstances may be maintained or shifted. However, the degree of interpersonal commitment is retained equally in general (90%). Consider example (3).

(3)
[ST]
不断　　[[推进]]　国家　　治理　　体系 和　治理　能力　现代化.
continually promote national governance system and capacity modernization
[TT]
further modernize China's national governance system and capacity.

 In the ST, *buduan*不断 (continually) is an adverb of frequency, and in the TT it was translated into an adverb of degree "further".

## Using prepositional phrases to achieve equal interpersonal commitment

Prepositional phrases tend to be employed to keep equal interpersonal commitment (95%), probably because phrases are usually experientially important in meanings. Example (4) is a case in point.

(4)
[ST]
以　　　可 持续　　发展　　　　[[促进]]　可　持续　安全 .
through sustainable development　promote sustainable security
[TT]
…and promote sustainable security through sustainable development.

 The phrase "through sustainable development" carries significant meanings both evaluatively and experientially. By the same token, other forms such as nouns and idiomatic phrases tend to maintain the same amount of interpersonal commitment, according to our observations.

## Using verbs to upscale interpersonal commitment

When verbs are chosen as the form for translation, more evaluative meanings along with experiential meanings tend to be activated (28%). Consider examples (5) and (6).

(5)
[ST]
…不断　[[推进]]　社会主义　现代化　建设.
continually promote socialist modernized construction
[TT]
.. and **endeavor to** advance the drive towards socialist modernization.

(6)
[ST]
长期以来 , 各　地区　各　部门　按照　中央　要求 , 不断　[[推进]]　公民道德　建设。
for a long time, every　local government　and every　department, to meet the central authorities' requirements,　continually promote the public morality construction.
[TT]
To meet the requirements of the central authorities, local governments and departments have **long worked hard to promote** public morality.

In example (5), the frequency adverb *buduan* 不断 (continually) was re-instantiated as a material process "endeavor to", forming a verbal group that "flags" (Martin & White 2015: 66) the attitude of determination and obligation. More obviously, in example (6), the adverb was shifted into "long worked hard to promote", a combination of duration and manner modifying the verb "work". It is clearly an upscaling of the material process to highlight the significance of "public morality".

**Using adjectives to upscale interpersonal commitment**

Adjectives are important resources for explicit evaluation. Thus, it is more likely that adjectives are used to raise interpersonal commitment in translation (31%). For example:

(7)
[ST]
不断　[[推进]]　制度　体系　完善　和　发展。
…continually promote the institution and system improving and developing.
[TT]
… and **be unstinting in our efforts to** improve and develop the institutional system.
(8)
[ST]
不久　将　全面　[[建成]]　小康社会,
soon will　comprehensively　build　a moderately prosperous society.
[TT]
… and will soon bring the building of a moderately prosperous society to a **successful** completion.

In example (7), the frequency adverb "*buduan* 不断 continually" was converted into the adjective "unstinting", explicitly manifesting the attitude of confidence. In example (8), the degree adverb "*quanmian* 全面 fully" was changed into the adjective "successful", indicating a judgment of capacity. More specifically, it is a case of inscribed appreciation and invoked judgement. "Completion", while it is a thing for the target of appreciation, is actually a nominalization of a verb, referring to the action of "complete", hence a judgment of how well the thing is done.

To sum up the major findings, translation choices of material processes are generally not sensitive to the occurrence of circumstances, which differs from previous findings on mental processes (Chen et al. 2023). Adverbs were the most frequent form to express circumstances in the ST, followed by prepositional phrases, idiomatic phrases, and complexes. In the TT, zero forms were the most common, accounting for 42% of the instances, indicating a significant reduction and a shift towards more simplified expressions. Certain grammatical forms in translation are associated with varying degrees of interpersonal commitment. Zero form generally represents a downscaling of commitment, while the use of verbs and adjectives tends to upscale it.

## Discussion

Based on the statistics above, we scrutinized each case involving mediation of interpersonal commitment and identified possible motivations underlying those translation choices.

The most obvious motivation is the reduction of rhetorical redundancy in translation for better communication effects. This motivation has been recognized in previous studies [16,18,25]. Cheng [18] argues that the repetitions of modifiers in Chinese political texts would sound verbose to English speakers, and thus should be carefully managed. This consideration can explain why omission of adverbs was the most frequent translation choice in our data. And we also found two rhetorical features from our data to support this line of reasoning.

The first feature is the four-character expressions used as circumstances. Four-character expressions are preferred in Chinese political discourse in general because of their conciseness and rhythmic quality. They can often evoke strong

emotions and resonance among the audience, thus enhancing the persuasiveness and mobilizing effects. English language, however, does not possess such expressive resources. In our data, there are 27 instances of idiomatic phrases, among which 25 instances involve four characters. The translation choices for those phrases are largely omission or reduction, hence a lower degree of interpersonal commitment. Consider the following examples:

(9)
[ST]
<u>坚定不移</u>　[[推进]]　理论　创新　…
resolutely without moving　promote　theoretical　innovation
[TT]
We must advance innovation in theory, …

(10)
[ST]
我们　要　<u>坚持不懈</u>　[[抓好]]　宪法　实施　工作　…
We　must　persistently without slack　grasp　the Constitution　implementation work
[TT]
We must <u>persistently</u> ensure the implementation of the Constitution…
Idiom phrase

In example (9), *jianding buyi* 坚定不移 is eliminated in translation. In example (10), *jianchi buxie* 坚持不懈 is turned into "persistently", whereby the rich connotation of determination of the phrase is lost in translation. The interpersonal commitment is weakened because the translation reduces the emphatic double-structure "坚持 (persist) + 不懈 (without slack)" into the single word "persistently." This loss of the reinforcing element "不懈" diminishes the original sense of unwavering, tenacious determination.

Another rhetorical feature that might motivate the down-tuning of interpersonal commitment is the adverbial complex. Our data show that some circumstances may contain a juxtaposition of two units such as "*qieshi youxiao* 切实有效 (solidly and effectively)", "*yibubu zhashi* 一步步扎实 (step by step and firmly)", etc. Similarly, the combination may have an enhancing and rhythmic effect but would probably be considered redundant in English. There are 16 instances of such cases (annotated as "complex") found in our data. Only 4 of them retain the complex structure in translation, the remaining being converted into single adverbs or other simpler forms with a lower degree of commitment. For example:

(11)
[ST]
…　切实有效　[[改善]]　民生　…
actually and effectively　improve　the people's livelihood
[TT]
… effectively raise living standards …

(12)
[ST]
…　一步步　扎实　向前　[[推进]]　…
step by step　firmly　forward　promote
[TT]
… implement it step by step.

In example (11), "*qieshi youxiao* 切实有效 (really and effectively)" is reduced to one single adverb "effectively" to avoid redundancy. In example (12), "*yibubu zhashi xiangqian* 一步步 扎实 向前 (step by step, firmly, forward)" is a triplet of three components. In translation, only "step by step" is re-instantiated with a loss of interpersonal commitment in "firmly" and "forward".

While previous research [18] well explains why interpersonal commitment is downscaled, few studies provide reasons why it is upscaled. Our data suggests that an upscaling of interpersonal commitment may show the "critical points" [11] of the translator's decision to emphasize a political agenda. Such instances are more likely to be found in two types of translation shifts in our data, namely, "adverb to adjective" and "adverb to verb". When circumstances are re-instantiated as adjectives, attitudes are more explicitly recognized in translation because adjectives are typical resources for expressing affect, appreciation, and judgment. Consider the following cases.

(13)
[ST]
…　稳步　[[推进]]　城镇化　健康　发展
steadily promote urbanization's healthy development
[TT]
… and promote <u>well-planned</u> and healthy urbanization.

In example (13), the adverb "*wenbu* 稳步 (steadily)" is converted to an adjective "well-planned", which serves as a modifier of the head "urbanization". Using terms from appraisal theory, the meaning of force has been turned into appreciation of "composition" [39], that is, attaching the value of balance to "urbanization".

Consider Example (8) above again. It reveals the translation decision to shift the adverb "*quanmian*全面 completely" to "successful completion", explicitly re-instantiating the meaning of degree into judgment, a subtype of attitude that points to behaviors. In brief, a conversion from adverbs to adjectives discloses the translation choices for a more explicit re-instantiation of the appraisal meaning, changing the interpersonal commitment from force to affect, appreciation or judgment. This shift of evaluative meaning reveals the translator's considerations to explicate the achievements of the major political plans such as urbanization and building of a prosperous society.

As for the translation shift from "adverb to verb", the most notable phenomenon is that various degrees of adverbs in Chinese have largely been re-instantiated as verbs of "working" and verbal phases of "making efforts". Among the 49 instances of "adverb to verb", 13 instances involve the lexical choice of "continue to"; 7 instances involve the lexical choice of "work to"; and 7 instances involve the choices of "effort" (see Table 6).

The most frequent shift is a translation from the adverb "*buduan* 不断 (continually)" to the verb "continue to", which still remains equal in interpersonal commitment. However, the second and the third frequent choices, "work (adverb) to" and "make (adjective) efforts to" often involve more interpersonal commitment.

(14)
[ST]
不断　[[推进]]　中国　特色　社会主义　制度　自我完善　和　发展 .
continually promote Chinese characteristic socialist system to improve and develop itself
[TT] and <u>work tirelessly to</u> promote the improvement and progress of socialism with Chinese characteristics.

(15)
[ST]
整体　[[推进]]　高校　党政干部　和　共青团　干部
comprehensively promote higher education institutions' Party teams and administrative officials
[TT]
We should <u>make an all-out effort to</u> build a high-quality team of Party and administrative officials in higher education institutions …

**Table 6. Lexical choices in translation shift from adverbs to verbs.**

| Items of shifts from adverbs to verbs | freq. | per. |
|---|---|---|
| continue to | 13 | 26.5% |
| work (adverb) to | 7 | 14.3% |
| make (adjective) efforts to | 7 | 14.3% |
| endeavor/strive to | 6 | 12.2% |
| make (adjective) moves | 2 | 4.1% |
| coordinate | 2 | 4.1% |
| commit to | 2 | 4.1% |
| … | 10 | 20.4% |
| Total | 49 | 100.0% |

Example (14) shows how interpersonal commitment is raised through converting "不断 continually" to "work tirelessly to", a more explicit judgment of "tenacity" (Martin & White [39]: 52). By the same token, in example (15), the adverb "*zhengti* 整体 comprehensively" is re-instantiated as "make an all-out effort", thus upscaling the amount of efforts to the maximum. All these instances demonstrate the translators' political consciousness on what should be evaluatively enhanced to construct the Party as a transforming agent in translation.

## Conclusion

This study has investigated the translation choices in political texts from Chinese to English, focusing on the mediation of interpersonal commitment to circumstances coupled with material processes. Employing the theoretical framework of re-instantiation within SFL, this research has provided a comprehensive description of how and why certain circumstances were shifted in translation with reference to grammatical forms and possible motivations based on data extracted from a corpus of institutional translation.

The findings reveal that while translation choices for material processes are generally not sensitive to the presence of circumstances, there is a notable shift in the expression of such circumstances. Adverbs were the most frequent form in the source text, yet zero forms were the most common in the translated text, indicating a significant reduction in the use of adverbs and a tendency towards simplified expressions. This suggests that translators often scale down interpersonal commitment to achieving acceptability and clarity in the target language. However, adjectives and verbs are more likely to be used to raise interpersonal commitment.

The study also identified some possible motivations behind the translation choices. Rhetorically, the avoidance of redundancy in English lead to a downscaling of interpersonal commitment in translation. Politically, the upscaling of commitment is used to emphasize efforts in institutional progress.

Overall, this research contributes to the understanding of how translation practices in the political domain are shaped by a complex interplay of linguistic, rhetorical, and political factors. It offers quantitative insights for both translation practitioners and scholars, emphasizing the usefulness of a linguistically minded corpus approach in political translation studies. It therefore paves the way for further exploration into the mediation of interpersonal meaning in other grammatical components or other discourse contexts.

## Supporting information

**S1 Data. Dataset.** Annotations for translation choices via Excel.
(XLSX)

## Author contributions

**Conceptualization:** Jiachun Li, Shukun Chen.

**Data curation:** Jiachun Li, Yawen Zhang.

**Formal analysis:** Jiachun Li, Shukun Chen.

**Funding acquisition:** Shukun Chen.

**Methodology:** Shukun Chen.

**Project administration:** Shukun Chen.

**Validation:** Yawen Zhang.

**Writing – original draft:** Jiachun Li, Shukun Chen.

**Writing – review & editing:** Yawen Zhang.

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
