## [Decision Letter · Decision Letter 0]

29 Apr 2025

PONE-D-25-02479“We must work tirelessly to promote …”: Mediating interpersonal commitment to material processes in English translation of Chinese political discoursePLOS ONE

Dear Dr. Chen,

Thank you for submitting your manuscript to PLOS ONE. After careful consideration, we feel that it has merit but does not fully meet PLOS ONE’s publication criteria as it currently stands. Therefore, we invite you to submit a revised version of the manuscript that addresses the points raised during the review process.

We look forward to receiving your revised manuscript.

Kind regards,

Fasih Ahmed

Academic Editor

PLOS ONE

Journal Requirements:

“Funder: Project of Guangdong Provincial Philosophy and Social Science Planning 2024

Project Title: A Study on the Projection Language Features in Political Discourse and Their Translation

Project Number: GD24CWY13”

3. We note that your Data Availability Statement is currently as follows: All relevant data are within the manuscript and in Supporting Information files.

4. Please ensure that you refer to Figure 1 in your text as, if accepted, production will need this reference to link the reader to the figure.

Additional Editor Comments:

Dear Author,

I have received the comments from reviewers and recommends major changes in the article. The concerns regarding the paper include, justification of the scale development, methodological limitations, Generalizability and Ethical Oversight.

Regards,

Fasih

Reviewers' comments:

Reviewer's Responses to Questions

**Comments to the Author**

1. Is the manuscript technically sound, and do the data support the conclusions?

Reviewer #1: Yes

Reviewer #2: Partly

Reviewer #3: Yes

2. Has the statistical analysis been performed appropriately and rigorously? 

Reviewer #1: Yes

Reviewer #2: Yes

Reviewer #3: Yes

3. Have the authors made all data underlying the findings in their manuscript fully available?

Reviewer #1: Yes

Reviewer #2: No

Reviewer #3: Yes

4. Is the manuscript presented in an intelligible fashion and written in standard English?

Reviewer #1: Yes

Reviewer #2: Yes

Reviewer #3: Yes

5. Review Comments to the Author

Reviewer #1: This paper examines the ways in which interpersonal commitment in the English translation of Chinese political discourse is mediated through evaluative items within appraisal system framework. While the topic is interesting and the method is proper, there are a couple of areas where the paper needs to be reworked in order to be publishable.

1. The authors need to provide more justification of corpus selection. While the corpus is authoritative, further discussion on potential biases or limitations in using Xi Jinping’s discourse as the primary dataset would increase the generalization of the research results.

2. The authors need to be aware of the potential international readers, not just Chinese readers. So when, for example, writing sth like “president Xi”, it might be more acceptable by using “Chinese president Xi Jinping” when it firstly is used.

3. Some sure it is reliable to make a certain argument. For example, “which differs from previous findings on mental processes”, here it would be helpful if some references are included.

4. Some sections contain very complex sentence structures. Please break them down, which is also related to comment 2 to take readers into account when writing English.

Basically, I would recommend it for publication after minor revision.

Reviewer #2: 1. Title Appropriateness

• The title is clear and indicates both the development and validation of a scale for digital resilience in adolescents. However:

o The term “digital resilience” is relatively novel and should be clearly conceptualized in the introduction to fully justify the use of the term.

o The context (e.g., cultural/geographic setting) is missing in the title. This limits the scope of understanding the applicability of the scale across populations.

2. Abstract

• The abstract lacks clarity regarding:

o The specific research gap this study addresses.

o The main findings of each phase (EFA/CFA) are not explicitly summarized.

o The implications are superficially mentioned without indicating the practical value of the scale for educators or policymakers.

o The methods are described in a general manner, without emphasizing the rationale behind the three phases.

3. Introduction

• The problem statement is vague—while digital risk and resilience are mentioned, the link between digital spaces and the need for a unique adolescent resilience scale is not strongly established.

• The literature gap is implied but not explicitly stated.

• The rationale for creating a new scale (rather than adapting or using existing ones) is not convincingly presented.

• The objectives could be more clearly listed for better coherence.

4. Literature Review

• Lacks critical engagement with prior scales measuring resilience—what specific limitations in those scales necessitate a new one?

• The concept of “digital resilience” is not adequately theorized using global or cross-cultural literature.

• More recent sources should be incorporated; some references seem dated for such a dynamic topic.

• The review doesn’t adequately support the dimensions later proposed in the scale.

5. Theoretical Framework

• There is no strong grounding in an established psychological or socio-cognitive theory of resilience.

• A diagrammatic representation or clear paragraph stating how theory underpins item generation would improve clarity.

• Lacks integration of developmental theories relevant to adolescents in digital contexts.

6. Methodology / Analysis

• Sampling: The justification for sample size and the geographic spread of participants is limited. Is the sample representative?

• Item generation: The process of item selection and reduction is insufficiently detailed. How were items retained or discarded?

• EFA/CFA: While the statistical techniques are correct, technical details (like factor loadings, KMO, Bartlett’s test, model fit indices) are not deeply interpreted.

• The cross-validation or test-retest reliability was not mentioned—this limits psychometric robustness.

7. Quantitative Findings

• Tables and figures are informative, but the narrative explanation is underdeveloped.

• Interpretation of factor analysis findings is superficial—e.g., why were these specific factors retained?

• Reliability statistics are reported but not critiqued—are they sufficient for real-world usage?

• The naming of factors appears intuitive but is not justified through strong theoretical or empirical linkage.

8. Discussion and Conclusion

• The discussion repeats results rather than offering deeper interpretative insight.

• There’s a lack of critical reflection on the limitations of the scale or how cultural/digital divides might affect generalizability.

• Implications are discussed broadly and lack concrete applications in digital education, policy, or adolescent psychology.

• The conclusion does not tie back well to the objectives or offer a vision for future research.

9. Academic Rigour

• While statistically sound, the conceptual and theoretical depth is weak.

• The study leans too heavily on quantitative methods without incorporating qualitative insights (e.g., interviews or expert evaluations).

• Lacks a section on ethical considerations for adolescent participants, especially when dealing with digital exposure.

10. References

• Some key references are missing—especially recent work on digital well-being and adolescent development.

• Over-reliance on a few sources to support multiple claims.

• Referencing style seems consistent but should be updated if the journal demands APA 7th or specific formatting.

Reviewer #3: This paper has investigated the translation choices in political texts from Chinese to English, focusing on the mediation of interpersonal commitment to circumstances coupled with material processes. Employing the theoretical framework of reinstantiation within Systemic Functional Linguistics (SFL), this research has provided a comprehensive description of how and why certain circumstances were shifted in translation, with reference to grammatical forms and possible motivations based on data extracted from a corpus of institutional translation.

The whole paper is divided into 7 parts. In Part One, the authors introduce the academic background of political translation. In Part Two, the authors review studies that have focused on political translation. In Part Three, the authors explain the theoretical framework of this paper. Part Four focuses on the methodology and presents the top ten verbs of material processes in Table One. Part Five, titled "Quantitative Findings," presents the results of this study. In this part, the authors not only present the research results but also analyze the translation methods. In Part Six, the authors discuss lexical choices in translation. In the end, the conclusion is drawn.

The presentation, writing skills, data and materials used, and the theoretical basis on which this paper relies are all rich and reliable. The data analysis is valuable and can be used.

This article has met the publication requirements.

Suggestions for Improvement:

It would be better if the authors provide several concrete examples in full sentences for Table 2 to prove the translation choices.

Examples from Ex. 2 lack concrete information, such as page numbers.

6. PLOS authors have the option to publish the peer review history of their article (what does this mean? ). If published, this will include your full peer review and any attached files.

**Do you want your identity to be public for this peer review?** For information about this choice, including consent withdrawal, please see our Privacy Policy .

Reviewer #1: **Yes: ** Tao Li

Reviewer #2: **Yes: ** Summiya Azam

Reviewer #3: No

---

## [Author Response · Author response to Decision Letter 1]

23 May 2025

Dear Professor Fasih Ahmed,

We have addressed all the issues raised in Journal Requirements and reviewer comments. However, please note that comments of Reviewer #2 seem to be irrelevant to our paper (PONE-D-25-02479) because it does not concern “digital resilience”. We therefore only responded to Reviewer #1 and Reviewer #3. Thank you very much!

Below are our responses marked in blue.

Journal Requirements:

//We have reformatted our manuscript to meet the style requirements.

“Funder: Project of Guangdong Provincial Philosophy and Social Science Planning 2024

Project Title: A Study on the Projection Language Features in Political Discourse and Their Translation

Project Number: GD24CWY13”

// We have added "The funders had no role in study design, data collection and analysis, decision to publish, or preparation of the manuscript." in the cover letter and manuscript. Thank you.

3. We note that your Data Availability Statement is currently as follows: All relevant data are within the manuscript and in Supporting Information files.

// We have provided a link to the data of our paper as below.

https://www.researchgate.net/publication/391812264_xijinpingtanzhiguolizhengyuliaobiao_advno_adv_20240315_LJC_20240317

We also added the following statement in the article

Data: The data of this study can be accessed through this link: https://www.researchgate.net/publication/391812264_xijinpingtanzhiguolizhengyuliaobiao_advno_adv_20240315_LJC_20240317

4. Please ensure that you refer to Figure 1 in your text as, if accepted, production will need this reference to link the reader to the figure.

// Thank you. We have referred to Figure 1 as follows:

We also refer to the Appraisal Theory (Martin & White 2005) to examine the interpersonal meaning in translation. In particular, we considered the system of graduation, as displayed in Figure 1. Graduation is one of the three main systems of meanings used to express positive or negative evaluations in discourse,. Graduation deals with adjusting the strength or intensity of an evaluative stance. It operates along two axes: force and focus.

Reviewer #1: This paper examines the ways in which interpersonal commitment in the English translation of Chinese political discourse is mediated through evaluative items within appraisal system framework. While the topic is interesting and the method is proper, there are a couple of areas where the paper needs to be reworked in order to be publishable.

1. The authors need to provide more justification of corpus selection. While the corpus is authoritative, further discussion on potential biases or limitations in using Xi Jinping’s discourse as the primary dataset would increase the generalization of the research results.

//Thank you. We have added the limitations of the data as follows

Despite its strengths, the corpus has limitations. First, the corpus is limited to Xi Jinping’s discourse, excluding other political figures or texts, which may restrict the generalizability of findings to broader Chinese political discourse. Second, manual annotation of grammatical forms and interpersonal commitment risks interpretive bias.

2. The authors need to be aware of the potential international readers, not just Chinese readers. So when, for example, writing sth like “president Xi”, it might be more acceptable by using “Chinese president Xi Jinping” when it firstly is used.

//Thanks. We have replaced president Xi with Chinese president Xi Jinping

3. Some sure it is reliable to make a certain argument. For example, “which differs from previous findings on mental processes”, here it would be helpful if some references are included.

// We have added the reference: Chen et al. 2023

4. Some sections contain very complex sentence structures. Please break them down, which is also related to comment 2 to take readers into account when writing English.

//Thank you. The complexity of some sentences might be due to theoretical thickness of the research design. We have tried to break down a few long sentences.

Basically, I would recommend it for publication after minor revision.

Reviewer #3: This paper has investigated the translation choices in political texts from Chinese to English, focusing on the mediation of interpersonal commitment to circumstances coupled with material processes. Employing the theoretical framework of reinstantiation within Systemic Functional Linguistics (SFL), this research has provided a comprehensive description of how and why certain circumstances were shifted in translation, with reference to grammatical forms and possible motivations based on data extracted from a corpus of institutional translation.

The whole paper is divided into 7 parts. In Part One, the authors introduce the academic background of political translation. In Part Two, the authors review studies that have focused on political translation. In Part Three, the authors explain the theoretical framework of this paper. Part Four focuses on the methodology and presents the top ten verbs of material processes in Table One. Part Five, titled "Quantitative Findings," presents the results of this study. In this part, the authors not only present the research results but also analyze the translation methods. In Part Six, the authors discuss lexical choices in translation. In the end, the conclusion is drawn.

The presentation, writing skills, data and materials used, and the theoretical basis on which this paper relies are all rich and reliable. The data analysis is valuable and can be used.

This article has met the publication requirements.

Suggestions for Improvement:

It would be better if the authors provide several concrete examples in full sentences for Table 2 to prove the translation choices.

// Thank you for your suggestion. Space does not allow us to provide full sentence examples. However, we offered a snapshot of our excel annotations to prove the translation choices. See Figure 2.

Examples from Ex. 2 lack concrete information, such as page numbers.

// Thanks. All examples are extracted from a corpus. So there is no information about page numbers.

Thank you very much for your helpful comments!

---

## [Editor Report · Decision Letter 1]

5 Jun 2025

PONE-D-25-02479R1“We must work tirelessly to promote …”: Mediating interpersonal commitment to material processes in English translation of Chinese political discoursePLOS ONE

Dear Dr. Chen,

Thank you for submitting your manuscript to PLOS ONE. After careful consideration, we feel that it has merit but does not fully meet PLOS ONE’s publication criteria as it currently stands. Therefore, we invite you to submit a revised version of the manuscript that addresses the points raised during the review process.

We look forward to receiving your revised manuscript.

Kind regards,

Fasih Ahmed

Academic Editor

PLOS ONE

Additional Editor Comments :

Dear Author,

After analyzing the commnents of the reviewers, I have reached to the decision of major revison.

Regards,

---

## [Author Response · Author response to Decision Letter 2]

5 Jun 2025

Dear Professor Fasih Ahmed,

We have addressed all the issues raised in Journal Requirements and reviewer comments. However, please note that comments of Reviewer #2 are not relevant to our paper (PONE-D-25-02479). We therefore only responded to Reviewer #1 and Reviewer #3. Thank you very much!

Below are our responses marked in blue.

Journal Requirements:

//We have reformatted our manuscript to meet the style requirements.

“Funder: Project of Guangdong Provincial Philosophy and Social Science Planning 2024

Project Title: A Study on the Projection Language Features in Political Discourse and Their Translation

Project Number: GD24CWY13”

// We have added "The funders had no role in study design, data collection and analysis, decision to publish, or preparation of the manuscript." in the cover letter and manuscript. Thank you.

3. We note that your Data Availability Statement is currently as follows: All relevant data are within the manuscript and in Supporting Information files.

// We have provided a link to the data of our paper as below.

https://www.researchgate.net/publication/391812264_xijinpingtanzhiguolizhengyuliaobiao_advno_adv_20240315_LJC_20240317

We also added the following statement in the article

Data: The data of this study can be accessed through this link: https://www.researchgate.net/publication/391812264_xijinpingtanzhiguolizhengyuliaobiao_advno_adv_20240315_LJC_20240317

4. Please ensure that you refer to Figure 1 in your text as, if accepted, production will need this reference to link the reader to the figure.

// Thank you. We have referred to Figure 1 as follows:

We also refer to the Appraisal Theory (Martin & White 2005) to examine the interpersonal meaning in translation. In particular, we considered the system of graduation, as displayed in Figure 1. Graduation is one of the three main systems of meanings used to express positive or negative evaluations in discourse,. Graduation deals with adjusting the strength or intensity of an evaluative stance. It operates along two axes: force and focus.

Reviewer #1: This paper examines the ways in which interpersonal commitment in the English translation of Chinese political discourse is mediated through evaluative items within appraisal system framework. While the topic is interesting and the method is proper, there are a couple of areas where the paper needs to be reworked in order to be publishable.

1. The authors need to provide more justification of corpus selection. While the corpus is authoritative, further discussion on potential biases or limitations in using Xi Jinping’s discourse as the primary dataset would increase the generalization of the research results.

//Thank you. We have added the limitations of the data as follows

Despite its strengths, the corpus has limitations. First, the corpus is limited to Xi Jinping’s discourse, excluding other political figures or texts, which may restrict the generalizability of findings to broader Chinese political discourse. Second, manual annotation of grammatical forms and interpersonal commitment risks interpretive bias.

2. The authors need to be aware of the potential international readers, not just Chinese readers. So when, for example, writing sth like “president Xi”, it might be more acceptable by using “Chinese president Xi Jinping” when it firstly is used.

//Thanks. We have replaced president Xi with Chinese president Xi Jinping

3. Some sure it is reliable to make a certain argument. For example, “which differs from previous findings on mental processes”, here it would be helpful if some references are included.

// We have added the reference: Chen et al. 2023

4. Some sections contain very complex sentence structures. Please break them down, which is also related to comment 2 to take readers into account when writing English.

//Thank you. The complexity of some sentences might be due to theoretical thickness of the research design. We have tried to break down a few long sentences.

Basically, I would recommend it for publication after minor revision.

Reviewer #3: This paper has investigated the translation choices in political texts from Chinese to English, focusing on the mediation of interpersonal commitment to circumstances coupled with material processes. Employing the theoretical framework of reinstantiation within Systemic Functional Linguistics (SFL), this research has provided a comprehensive description of how and why certain circumstances were shifted in translation, with reference to grammatical forms and possible motivations based on data extracted from a corpus of institutional translation.

The whole paper is divided into 7 parts. In Part One, the authors introduce the academic background of political translation. In Part Two, the authors review studies that have focused on political translation. In Part Three, the authors explain the theoretical framework of this paper. Part Four focuses on the methodology and presents the top ten verbs of material processes in Table One. Part Five, titled "Quantitative Findings," presents the results of this study. In this part, the authors not only present the research results but also analyze the translation methods. In Part Six, the authors discuss lexical choices in translation. In the end, the conclusion is drawn.

The presentation, writing skills, data and materials used, and the theoretical basis on which this paper relies are all rich and reliable. The data analysis is valuable and can be used.

This article has met the publication requirements.

Suggestions for Improvement:

It would be better if the authors provide several concrete examples in full sentences for Table 2 to prove the translation choices.

// Thank you for your suggestion. Space does not allow us to provide full sentence examples. However, we offered a snapshot of our excel annotations to prove the translation choices. See Figure 2.

Examples from Ex. 2 lack concrete information, such as page numbers.

// Thanks. All examples are extracted from a corpus. So there is no information about page numbers.

Thank you very much for your helpful comments!

---

## [Decision Letter · Decision Letter 2]

23 Oct 2025

PONE-D-25-02479R2“We must work tirelessly to promote …”: Mediating interpersonal commitment to material processes in English translation of Chinese political discoursePLOS ONE

Dear Dr. Chen,

Thank you for submitting your manuscript to PLOS ONE. After careful consideration, we feel that it has merit but does not fully meet PLOS ONE’s publication criteria as it currently stands. Therefore, we invite you to submit a revised version of the manuscript that addresses the points raised during the review process.

We look forward to receiving your revised manuscript.

Kind regards,

Awni Etaywe, PhD

Academic Editor

PLOS ONE

Journal Requirements:

Additional Editor Comments (if provided):

Dear Authors,

Thank you for submitting your manuscript to PLOS ONE and for your thoughtful work on this topic. Both reviewers found your study to be a valuable contribution to the field and well-aligned with the journal’s scope. They have recommended minor revisions to improve clarity and precision in presentation.

Based on the reviewers’ feedback and my own assessment, I concur that your manuscript requires only minor adjustments before it can be considered for acceptance. Please address each of the reviewers’ comments point by point and include a revised version of the manuscript with changes clearly marked.

Please provide a detailed response letter outlining how each comment was addressed and indicating where revisions have been made in the manuscript.

Once these minor revisions are incorporated, your manuscript should be suitable for publication.

Kind regards,

Reviewers' comments:

Reviewer's Responses to Questions

**Comments to the Author**

1. If the authors have adequately addressed your comments raised in a previous round of review and you feel that this manuscript is now acceptable for publication, you may indicate that here to bypass the “Comments to the Author” section, enter your conflict of interest statement in the “Confidential to Editor” section, and submit your "Accept" recommendation.

Reviewer #4: (No Response)

Reviewer #5: All comments have been addressed

2. Is the manuscript technically sound, and do the data support the conclusions?

Reviewer #4: Yes

Reviewer #5: Yes

3. Has the statistical analysis been performed appropriately and rigorously? 

Reviewer #4: N/A

Reviewer #5: Yes

4. Have the authors made all data underlying the findings in their manuscript fully available?

Reviewer #4: Yes

Reviewer #5: Yes

5. Is the manuscript presented in an intelligible fashion and written in standard English?

Reviewer #4: Yes

Reviewer #5: Yes

6. Review Comments to the Author

Reviewer #4: This manuscript touches on a very interesting topic. The selected political texts provide a good source for investigating the shifts of interpersonal meaning between the Chinese original text (ST) and its English translated text (TT). SFL is a systemic approach to analyse how interpersonal meaning is instantiated in the ST and how the meaning is re-instantiated in the TT. The methodology shows that, by focusing on the pattern of “circumstances + material processes”, the authors had made a scrutinized examination of the translation shifts in grammatical forms. The findings provide a good foundation for further investigation into the potential translation shifts in terms of interpersonal commitment. Therefore, the overall design of this project is quite promising.

The manuscript is full of potential. However, a few issues need to be addressed before publication.

1. Page 5, “Prescriptive studies offer detailed description of what really happened in political

translation, mostly relying on a corpus method.”

There is a typo; this should be an explanation of ‘Descriptive studies’. Furthermore, after introducing the prescriptive and descriptive approaches, it is better to clarify which approach this project takes.

2. Page 5-6, “More specifically, the instantiation cline involves system, genre/register, text type, text, and reading, where system represents the most general end of meaning potential and reading stands for subjectified meaning, the most actual end.”

This may be a bit hard to follow for readers from a non-SFL background. Ideally, it is better to break it down a bit for the readers. The authors may consider, at least, providing a figure of the instantiation cline and directing the readers to a relevant reference. The authors can use the one in Martin and White (2005) or another work they prefer.

3. Page 11, “The top five circumstances are buduan 不断 (continually), quanmian 全面 (fully), jiji 积极 (actively), xietiao 协调 (coordinately), and qieshi 切实 (practically)…These circumstances boost the force of the material processes, thus raising the mobilizing effects of the political communication.”

The manuscript states that all circumstances included in this project refer to an instance of intensification. From the perspective of Appraisal, it seems hard to take 协调 (coordinately) and qieshi 切实 (practically) as instances of intensification. Would the authors please clarify the reasons for identifying the two items as ‘GRADUATION: intensification’?

3. Page 14, Example (4), under the heading of “Using prepositional phrases to achieve equal interpersonal commitment”:

[ST]

以 更加 开放 的 胸襟 和 更加 积极 的 态度 [[促进]] 地区 合作 .

with an opener mind and more positive attitude promote regional cooperation

[TT]

… and promoting regional cooperation with an open mind and enthusiasm.

Comparatives are commonly used as intensifiers; they usually indicate an upscaling of intensity or of quantity. In this example, the two comparatives in the ST were omitted in the TT, this would have an impact on the rhetorical force and eventually affect the interpersonal commitment. Would the authors please clarify the reason why this example was taken as achieving an equal interpersonal commitment?

4. Page 15, Example (8):

[ST]

不久 将 全面 [[建成]] 小康社会,

soon will comprehensively build a moderately prosperous society.

[TT]

… and will soon bring the building of a moderately prosperous society to a successful completion.

“…In example (8), the degree adverb “quanmian 全面 fully” was changed into the adjective ‘successful’, indicating a judgment of capacity.”

In the English text – ‘a successful completion’ – completion as the target of this evaluation refers to a thing rather than a behaviour.

5. Page 17, Example (10):

[ST]

我们 要 坚持不懈 [[抓好]] 宪法 实施 工作 …

We must persistently without slack grasp the Constitution implementation work

[TT]

We must persistently ensure the implementation of the Constitution … Idiom phrase

“…In example (10), jianchi buxie 坚持不懈 is turned into “persistently”, whereby the rich

connotation of determination of the phrase is lost in translation.”

The authors will need to elaborate on the explanation for this instance, because at the semantic level there seems to be no shift.

6. Page 19, Example (13):

[ST]

… 稳步 [[推进]] 城镇化 健康 发展

steadily promote urbanization’s healthy development

[TT]

… and promote well-planned and healthy urbanization.

“In example (13), the adverb ‘wenbu 稳步 (steadily)’ is converted to an adjective ‘well-planned’, which serves as a modifier of the head ‘urbanization’. Using terms from appraisal theory, the meaning of force has been turned into appreciation of ‘composition’ [39], that is, attaching the value of balance to ‘urbanization’.”

Would the authors please clarify the reason for identifying wenbu 稳步 (steadily) as an instance of ‘GRADUATION: intensification’?

Reviewer #5: Please find my comments attached in the pdf. I think you revised your paper according to the reviewers comments properly. However, I have provided you with further comments to improve the quality of your work

7. PLOS authors have the option to publish the peer review history of their article (what does this mean? ). If published, this will include your full peer review and any attached files.

**Do you want your identity to be public for this peer review?** For information about this choice, including consent withdrawal, please see our Privacy Policy .

Reviewer #4: No

Reviewer #5: No

---

## [Author Response · Author response to Decision Letter 3]

28 Oct 2025

Dear Editor,

We are grateful for the helpful comment of the reviewer. We have addressed each issue raised by the reviewer. Please see below for our response.

This manuscript touches on a very interesting topic. The selected political texts provide a good source for investigating the shifts of interpersonal meaning between the Chinese original text (ST) and its English translated text (TT). SFL is a systemic approach to analyse how interpersonal meaning is instantiated in the ST and how the meaning is re-instantiated in the TT. The methodology shows that, by focusing on the pattern of “circumstances + material processes”, the authors had made a scrutinized examination of the translation shifts in grammatical forms. The findings provide a good foundation for further investigation into the potential translation shifts in terms of interpersonal commitment. Therefore, the overall design of this project is quite promising.

The manuscript is full of potential. However, a few issues need to be addressed before publication.

1. Page 5, “Prescriptive studies offer detailed description of what really happened in political

translation, mostly relying on a corpus method.”

There is a typo; this should be an explanation of ‘Descriptive studies’. Furthermore, after introducing the prescriptive and descriptive approaches, it is better to clarify which approach this project takes.

//Response: Thank you for pointing that out. We have corrected the typo and clarify the approach of this study as follows:

Descriptive studies offer detailed description of what really happened in political translation, mostly relying on a corpus method. The data generated out of a large-scale corpus provide insights into linguistic features of various political texts and patterns of translation choices [26-28]. Translation studies of interpersonal resources have been particularly fruitful in this respect [7, 8, 11, 12, 13, 14]. Yu and Wang [7] find that in Chinese-to-English translation, high-valued modal expressions in the source texts are often omitted or translated into lower-valued expressions, while median-valued modal expressions tend to be retained. Additionally, the translated texts exhibit a shift towards a more target-oriented tone aligned with English communicative conventions and the intended persuasive function in the target culture. This study adopts exactly this descriptive approach to examine interpersonal shifts in political translation.

2. Page 5-6, “More specifically, the instantiation cline involves system, genre/register, text type, text, and reading, where system represents the most general end of meaning potential and reading stands for subjectified meaning, the most actual end.”

This may be a bit hard to follow for readers from a non-SFL background. Ideally, it is better to break it down a bit for the readers. The authors may consider, at least, providing a figure of the instantiation cline and directing the readers to a relevant reference. The authors can use the one in Martin and White (2005) or another work they prefer.

//Response: Thank you. We have added the following figure to illustrate the instantiation cline.

Figure 1. Instantiation as a cline from reading to system (Martin [30])

3. Page 11, “The top five circumstances are buduan 不断 (continually), quanmian 全面 (fully), jiji 积极 (actively), xietiao 协调 (coordinately), and qieshi 切实 (practically)…These circumstances boost the force of the material processes, thus raising the mobilizing effects of the political communication.”

The manuscript states that all circumstances included in this project refer to an instance of intensification. From the perspective of Appraisal, it seems hard to take 协调 (coordinately) and qieshi 切实 (practically) as instances of intensification. Would the authors please clarify the reasons for identifying the two items as ‘GRADUATION: intensification’?

//Response: Thank you. This is indeed a good question. Actually, we included circumstances with evaluative sense of frequency, degree and manner, going beyond the scope of ‘GRADUATION: intensification’. We have corrected the relevant parts to avoid confusion as follows:

In the case of the current study, we aim to observe how the meaning of force has come to enhance the material processes in Chinese political texts, and the correspondent translation choices. That means, only those circumstances that carry the meaning of evaluative force (i.e. degree, frequency and manner) are included for the current study.

3. Page 14, Example (4), under the heading of “Using prepositional phrases to achieve equal interpersonal commitment”:

[ST]

以 更加 开放 的 胸襟 和 更加 积极 的 态度 [[促进]] 地区 合作 .

with an opener mind and more positive attitude promote regional cooperation

[TT]

… and promoting regional cooperation with an open mind and enthusiasm.

Comparatives are commonly used as intensifiers; they usually indicate an upscaling of intensity or of quantity. In this example, the two comparatives in the ST were omitted in the TT, this would have an impact on the rhetorical force and eventually affect the interpersonal commitment. Would the authors please clarify the reason why this example was taken as achieving an equal interpersonal commitment?

//Response:

Thank you. This is a mistake. We checked our annotation in our dataset. This is indeed a case of evaluative downgrade. So we changed the example as follows:

4. Page 15, Example (8):

[ST]

不久 将 全面 [[建成]] 小康社会,

soon will comprehensively build a moderately prosperous society.

[TT]

… and will soon bring the building of a moderately prosperous society to a successful completion.

“…In example (8), the degree adverb “quanmian 全面 fully” was changed into the adjective ‘successful’, indicating a judgment of capacity.”

In the English text – ‘a successful completion’ – completion as the target of this evaluation refers to a thing rather than a behaviour.

//Response:

This is a case of grammatical metaphor. Completion, while it is a thing, is actually a nominalization of a verb, referring to the action of “complete”, hence a judgment of how well the thing is done.

5. Page 17, Example (10):

[ST]

我们 要 坚持不懈 [[抓好]] 宪法 实施 工作 …

We must persistently without slack grasp the Constitution implementation work

[TT]

We must persistently ensure the implementation of the Constitution … Idiom phrase

“…In example (10), jianchi buxie 坚持不懈 is turned into “persistently”, whereby the rich

connotation of determination of the phrase is lost in translation.”

The authors will need to elaborate on the explanation for this instance, because at the semantic level there seems to be no shift.

//Response: Thank you. We explained this as follows:

The interpersonal commitment is weakened because the translation reduces the emphatic double-structure "坚持 (persist) + 不懈 (without slack)" into the single word "persistently." This loss of the reinforcing element "不懈" diminishes the original sense of unwavering, tenacious determination.

6. Page 19, Example (13):

[ST]

… 稳步 [[推进]] 城镇化 健康 发展

steadily promote urbanization’s healthy development

[TT]

… and promote well-planned and healthy urbanization.

“In example (13), the adverb ‘wenbu 稳步 (steadily)’ is converted to an adjective ‘well-planned’, which serves as a modifier of the head ‘urbanization’. Using terms from appraisal theory, the meaning of force has been turned into appreciation of ‘composition’ [39], that is, attaching the value of balance to ‘urbanization’.”

Would the authors please clarify the reason for identifying wenbu 稳步 (steadily) as an instance of ‘GRADUATION: intensification’?

//Response:

wenbu 稳步 (steadily) is a case of manner, not intensification. See our correction above. Thank you!

Many thanks again for the correction and insightful comment.

---

## [Decision Letter · Decision Letter 3]

5 Nov 2025

PONE-D-25-02479R3“We must work tirelessly to promote …”: Mediating interpersonal commitment to material processes in English translation of Chinese political discoursePLOS ONE

Dear Dr. Chen,

Thank you for submitting your manuscript to PLOS ONE. After careful consideration, we feel that it has merit but does not fully meet PLOS ONE’s publication criteria as it currently stands. Therefore, we invite you to submit a revised version of the manuscript that addresses the points raised during the review process.

We look forward to receiving your revised manuscript.

Kind regards,

Awni Etaywe, PhD

Academic Editor

PLOS ONE

Journal Requirements:

Additional Editor Comments (if provided):

Some minor revisions as suggested by Reviewer 1 still need to be responded to and integrated into the paper itself.

Reviewers' comments:

Reviewer's Responses to Questions

**Comments to the Author**

1. If the authors have adequately addressed your comments raised in a previous round of review and you feel that this manuscript is now acceptable for publication, you may indicate that here to bypass the “Comments to the Author” section, enter your conflict of interest statement in the “Confidential to Editor” section, and submit your "Accept" recommendation.

Reviewer #4: (No Response)

2. Is the manuscript technically sound, and do the data support the conclusions?

Reviewer #4: Yes

3. Has the statistical analysis been performed appropriately and rigorously? 

Reviewer #4: N/A

4. Have the authors made all data underlying the findings in their manuscript fully available?

Reviewer #4: Yes

5. Is the manuscript presented in an intelligible fashion and written in standard English?

Reviewer #4: Yes

6. Review Comments to the Author

Reviewer #4: Page 15, Example (8):

[ST]

不久 将 全面 [[建成]] 小康社会,

soon will comprehensively build a moderately prosperous society.

[TT]

… and will soon bring the building of a moderately prosperous society to a successful completion.

“…In example (8), the degree adverb “quanmian 全面 fully” was changed into the

adjective ‘successful’, indicating a judgment of capacity.”

In the English text – ‘a successful completion’ – completion as the target of this

evaluation refers to a thing rather than a behaviour.

//Response:

This is a case of grammatical metaphor. Completion, while it is a thing, is actually a nominalization of a verb, referring to the action of “complete”, hence a judgment of how

well the thing is done.

- Yes, we can discuss this case as inscribed appreciation & invoked judgement. The authors may consider providing the above clarification for the reader.

Page 11, “The top five circumstances are buduan 不断 (continually), quanmian 全面 (fully), jiji 积极 (actively), xietiao 协调 (coordinately), and qieshi 切实 (practically)…These circumstances boost the force of the material processes, thus raising the mobilizing effects of the political communication.” The manuscript states that all circumstances included in this project refer to an instance of intensification. From the perspective of Appraisal, it seems hard to take 协调 (coordinately) and qieshi 切实 (practically) as instances of intensification. Would the authors please clarify the reasons for identifying the two items as ‘GRADUATION:

intensification’?

//Response: Thank you. This is indeed a good question. Actually, we included circumstances with evaluative sense of frequency, degree and manner, going beyond the scope of ‘GRADUATION: intensification’. We have corrected the relevant parts to avoid confusion as follows:

In the case of the current study, we aim to observe how the meaning of force has come to enhance the material processes in Chinese political texts, and the correspondent translation choices. That means, only those circumstances that carry the meaning of evaluative force (i.e. degree, frequency and manner) are included for the current study.

- The term ‘intensification’ in the GRADUATION system may cause some confusion. It refers to the author’s up/down scaling of qualities and processes, and manner is an important resource. For this case, the attitudinal meanings in 协调 (coordinately) and 切实 (practically) are quite obvious. However, my concern is whether these two items indicate a subjective scaling.

Page 19, Example (13):

[ST]

… 稳步 [[推进]] 城镇化 健康 发展

steadily promote urbanization’s healthy development

[TT]

… and promote well-planned and healthy urbanization.

“In example (13), the adverb ‘wenbu 稳步 (steadily)’ is converted to an adjective ‘wellplanned’,

which serves as a modifier of the head ‘urbanization’. Using terms from appraisal theory, the meaning of force has been turned into appreciation of ‘composition’ [39], that is, attaching the value of balance to ‘urbanization’.” Would the authors please clarify the reason for identifying wenbu 稳步 (steadily) as an instance of ‘GRADUATION: intensification’?

//Response:

wenbu 稳步 (steadily) is a case of manner, not intensification. See our correction above.

- The same concern as above.

7. PLOS authors have the option to publish the peer review history of their article (what does this mean? ). If published, this will include your full peer review and any attached files.

**Do you want your identity to be public for this peer review?** For information about this choice, including consent withdrawal, please see our Privacy Policy .

Reviewer #4: No

---

## [Author Response · Author response to Decision Letter 4]

8 Nov 2025

Dear Editor,

We are grateful for your meticulous work. Please see below for our response.

//Response:

One reviewer recommended to cite a previously published work to indicate how findings of the current study differed from previous ones (Chen et al. 2023). We have evaluated it and confirmed that it is necessary.

//Response:

We have reviewed the reference list and confirmed that it is complete and correct.

Additional Editor Comments (if provided):

Some minor revisions as suggested by Reviewer 1 still need to be responded to and integrated into the paper itself.

Reviewer #4: Page 15, Example (8):

[ST]

不久 将 全面 [[建成]] 小康社会,

soon will comprehensively build a moderately prosperous society.

[TT]

… and will soon bring the building of a moderately prosperous society to a successful completion.

“…In example (8), the degree adverb “quanmian 全面 fully” was changed into the

adjective ‘successful’, indicating a judgment of capacity.”

In the English text – ‘a successful completion’ – completion as the target of this

evaluation refers to a thing rather than a behaviour.

//Response:

This is a case of grammatical metaphor. Completion, while it is a thing, is actually a nominalization of a verb, referring to the action of “complete”, hence a judgment of how well the thing is done.

- Yes, we can discuss this case as inscribed appreciation & invoked judgement. The authors may consider providing the above clarification for the reader.

//Response:

Thank you. It is indeed more accurate to use inscribed appreciation & invoked judgement to describe the phenomenon. We have added the discussion in the article as follows

More specifically, it is a case of inscribed appreciation and invoked judgement. “Completion”, while it is a thing for the target of appreciation, is actually a nominalization of a verb, referring to the action of “complete”, hence a judgment of how well the thing is done.

Page 11, “The top five circumstances are buduan 不断 (continually), quanmian 全面 (fully), jiji 积极 (actively), xietiao 协调 (coordinately), and qieshi 切实 (practically)…These circumstances boost the force of the material processes, thus raising the mobilizing effects of the political communication.” The manuscript states that all circumstances included in this project refer to an instance of intensification. From the perspective of Appraisal, it seems hard to take 协调 (coordinately) and qieshi 切实 (practically) as instances of intensification. Would the authors please clarify the reasons for identifying the two items as ‘GRADUATION:

intensification’?

//Response: Thank you. This is indeed a good question. Actually, we included circumstances with evaluative sense of frequency, degree and manner, going beyond the scope of ‘GRADUATION: intensification’. We have corrected the relevant parts to avoid confusion as follows:

In the case of the current study, we aim to observe how the meaning of force has come to enhance the material processes in Chinese political texts, and the correspondent translation choices. That means, only those circumstances that carry the meaning of evaluative force (i.e. degree, frequency and manner) are included for the current study.

- The term ‘intensification’ in the GRADUATION system may cause some confusion. It refers to the author’s up/down scaling of qualities and processes, and manner is an important resource. For this case, the attitudinal meanings in 协调 (coordinately) and 切实 (practically) are quite obvious. However, my concern is whether these two items indicate a subjective scaling.

//Response:

You are right to point out that 协调 (coordinately) and 切实 (practically) do not necessarily intensify the verb in terms of degree. However, our research design deliberately sets out to track any evaluative adverb—regardless of whether it inherently scales the process up or down—as long as it carries attitudinal meaning. In other words, some of the adverbs involve scaling of the process, some might not. But we counted all of them as long as they have evaluative meanings. And we wanted to observe how these evaluative meanings are upscaled or downscaled in translation in terms of interpersonal commitment.

The term ‘intensification’ in the GRADUATION system may cause some confusion.

//Response:

The term ‘intensification’ was replaced by ‘evaluative force’ to avoid confusion.

Page 19, Example (13):

[ST]

… 稳步 [[推进]] 城镇化 健康 发展

steadily promote urbanization’s healthy development

[TT]

… and promote well-planned and healthy urbanization.

“In example (13), the adverb ‘wenbu 稳步 (steadily)’ is converted to an adjective ‘wellplanned’,

which serves as a modifier of the head ‘urbanization’. Using terms from appraisal theory, the meaning of force has been turned into appreciation of ‘composition’ [39], that is, attaching the value of balance to ‘urbanization’.” Would the authors please clarify the reason for identifying wenbu 稳步 (steadily) as an instance of ‘GRADUATION: intensification’?

//Response:

wenbu 稳步 (steadily) is a case of manner, not intensification. See our correction above.

- The same concern as above.

//Response:

Thank you for the concern. Whether wenbu 稳步 (steadily) involves subjective scaling does not impact our findings because this study aims to observe the translation shifts of the adverbs containing evaluative meanings. 稳步 (steadily) has a meaning of “steady” and “moderate”. That is why we included this item for analysis for translation shifts.

Many thanks again for Reviewer’s comments and discussion. They are helpful for improvement of the paper.

---

## [Decision Letter · Decision Letter 4]

16 Nov 2025

PONE-D-25-02479R4“We must work tirelessly to promote …”: Mediating interpersonal commitment to material processes in English translation of Chinese political discoursePLOS ONE

Dear Dr. Chen,

Thank you for submitting your manuscript to PLOS ONE. After careful consideration, we feel that it has merit but does not fully meet PLOS ONE’s publication criteria as it currently stands. Therefore, we invite you to submit a revised version of the manuscript that addresses the points raised during the review process.

We look forward to receiving your revised manuscript.

Kind regards,

Awni Etaywe, PhD

Academic Editor

PLOS ONE

Journal Requirements:

Additional Editor Comments (if provided):

Dear Authors,

The manuscript presents a clearly structured study with sound methodology and results that are relevant to the field. The contribution is valuable and meets PLOS ONE’s criteria for technical rigor and clarity. The paper is publishable; only minor revisions are needed to strengthen the clarity and transparency of the article, particularly in its focus on the Appraisal system—should the authors incorporate the reviewers’ suggestions

Overall, I recommend Minor Revision. The manuscript will be suitable for publication once the authors address the reviewers’ comments.

Best Regards,

Dr Awni Etaywe

Reviewers' comments:

Reviewer's Responses to Questions

**Comments to the Author**

1. If the authors have adequately addressed your comments raised in a previous round of review and you feel that this manuscript is now acceptable for publication, you may indicate that here to bypass the “Comments to the Author” section, enter your conflict of interest statement in the “Confidential to Editor” section, and submit your "Accept" recommendation.

Reviewer #4: (No Response)

2. Is the manuscript technically sound, and do the data support the conclusions?

Reviewer #4: Yes

3. Has the statistical analysis been performed appropriately and rigorously? 

Reviewer #4: Yes

4. Have the authors made all data underlying the findings in their manuscript fully available?

Reviewer #4: Yes

5. Is the manuscript presented in an intelligible fashion and written in standard English?

Reviewer #4: Yes

6. Review Comments to the Author

Reviewer #4: Page 11, “The top five circumstances are buduan 不断 (continually), quanmian 全面 (fully), jiji 积极 (actively), xietiao 协调 (coordinately), and qieshi 切实(practically)…These circumstances boost the force of the material processes, thus raising the mobilizing effects of the political communication.” The manuscript states that all circumstances included in this project refer to an instance of intensification.

From the perspective of Appraisal, it seems hard to take 协调 (coordinately) and qieshi切实 (practically) as instances of intensification. Would the authors please clarify the reasons for identifying the two items as ‘GRADUATION:intensification’?

//Response: Thank you. This is indeed a good question. Actually, we included circumstances with evaluative sense of frequency, degree and manner, going beyond the scope of ‘GRADUATION: intensification’. We have corrected the relevant parts to avoid confusion as follows:

In the case of the current study, we aim to observe how the meaning of force has come to enhance the material processes in Chinese political texts, and the correspondent translation choices. That means, only those circumstances that carry the meaning of evaluative force (i.e. degree, frequency and manner) are included for the current study.

- The term ‘intensification’ in the GRADUATION system may cause some confusion. It refers to the author’s up/down scaling of qualities and processes, and manner is an important resource. For this case, the attitudinal meanings in 协调 (coordinately) and切实 (practically) are quite obvious. However, my concern is whether these two items indicate a subjective scaling.

//Response:

You are right to point out that 协调 (coordinately) and 切实 (practically) do not necessarily intensify the verb in terms of degree. However, our research design deliberately sets out to track any evaluative adverb—regardless of whether it inherently scales the process up or down—as long as it carries attitudinal meaning. In other words, some of the adverbs involve scaling of the process, some might not. But we counted all of them as long as they have evaluative meanings. And we wanted to observe how these evaluative meanings are upscaled or downscaled in translation in terms of interpersonal commitment.

- Thanks for the clarification. I would recommend the authors clarify in the ‘theoretical framework’ that the analyses of this paper were concerned with GRADUATION and ATTITUDE. This will be my last comment. I have no further concerns.

7. PLOS authors have the option to publish the peer review history of their article (what does this mean? ). If published, this will include your full peer review and any attached files.

**Do you want your identity to be public for this peer review?** For information about this choice, including consent withdrawal, please see our Privacy Policy .

Reviewer #4: No

---

## [Author Response · Author response to Decision Letter 5]

17 Nov 2025

Dear Editor,

We are grateful for your meticulous work. Please see below for our response.

Reviewer #4: Page 11, “The top five circumstances are buduan 不断 (continually), quanmian 全面 (fully), jiji 积极 (actively), xietiao 协调 (coordinately), and qieshi 切实(practically)…These circumstances boost the force of the material processes, thus raising the mobilizing effects of the political communication.” The manuscript states that all circumstances included in this project refer to an instance of intensification.

From the perspective of Appraisal, it seems hard to take 协调 (coordinately) and qieshi切实 (practically) as instances of intensification. Would the authors please clarify the reasons for identifying the two items as ‘GRADUATION:intensification’?

//Response: Thank you. This is indeed a good question. Actually, we included circumstances with evaluative sense of frequency, degree and manner, going beyond the scope of ‘GRADUATION: intensification’. We have corrected the relevant parts to avoid confusion as follows:

In the case of the current study, we aim to observe how the meaning of force has come to enhance the material processes in Chinese political texts, and the correspondent translation choices. That means, only those circumstances that carry the meaning of evaluative force (i.e. degree, frequency and manner) are included for the current study.

- The term ‘intensification’ in the GRADUATION system may cause some confusion. It refers to the author’s up/down scaling of qualities and processes, and manner is an important resource. For this case, the attitudinal meanings in 协调 (coordinately) and切实 (practically) are quite obvious. However, my concern is whether these two items indicate a subjective scaling.

//Response:

You are right to point out that 协调 (coordinately) and 切实 (practically) do not necessarily intensify the verb in terms of degree. However, our research design deliberately sets out to track any evaluative adverb—regardless of whether it inherently scales the process up or down—as long as it carries attitudinal meaning. In other words, some of the adverbs involve scaling of the process, some might not. But we counted all of them as long as they have evaluative meanings. And we wanted to observe how these evaluative meanings are upscaled or downscaled in translation in terms of interpersonal commitment.

- Thanks for the clarification. I would recommend the authors clarify in the ‘theoretical framework’ that the analyses of this paper were concerned with GRADUATION and ATTITUDE. This will be my last comment. I have no further concerns.

//Response:

Thank you. We have added a few words and a paragraph to clarify our research concerns as follows:

We also refer to the Appraisal Theory [39] to examine the interpersonal meaning in translation. In particular, we considered the system of graduation and attitude, as displayed in Fig 2. Graduation is one of the three main systems of meanings used to express positive or negative evaluations in discourse. Graduation deals with adjusting the strength or intensity of an evaluative stance. It operates along two axes: force and focus.

Force concerns the intensity or amount of the evaluated phenomenon. Force can be either "raised" (upscaled) or "lowered" (downscaled). For example, the words "very," "extremely," or "somewhat" can be used to modify the force of an adjective or adverb.

Focus sharpens or blurs the boundaries of the evaluative term. Under focus, we can distinguish between "sharpening" (specifying a term more precisely) and "softening" (vagueness or imprecision). For instance, words like "real" or "true" can sharpen the focus of a category, while "sort of" or "kind of" can soften it.

The Attitude subsystem represents choices of expressing feelings and evaluations. It is divided into three core domains: Affect, which deals with emotional responses and feelings; Judgement, which concerns the ethical assessment of behavior and character; and Appreciation, which evaluates the aesthetic and social value of objects, texts, and phenomena.

In the case of the current study, we aim to observe how the meaning of force and attitude has come to enhance the material processes in Chinese political texts, and the correspondent translation choices. That means, only those circumstances that carry the meaning of evaluative force (i.e. degree, frequency and manner) are included for the current study.

Many thanks again for Reviewer’s comments and discussion. They are helpful for improvement of the paper.

---

## [Decision Letter · Decision Letter 5]

23 Nov 2025

“We must work tirelessly to promote …”: Mediating interpersonal commitment to material processes in English translation of Chinese political discourse

PONE-D-25-02479R5

Dear Dr. Chen,

We’re pleased to inform you that your manuscript has been judged scientifically suitable for publication and will be formally accepted for publication once it meets all outstanding technical requirements.

Kind regards,

Awni Etaywe, PhD

Academic Editor

PLOS ONE

Additional Editor Comments (optional):

Dear Authors,

Thank you for your thoughtful and comprehensive revisions to the manuscript, “We must work tirelessly to promote …”: Mediating interpersonal commitment to material processes in English translation of Chinese political discourse. It is clear that you have carefully addressed each of the reviewers’ comments, providing clarifications where needed and strengthening the theoretical framing, particularly in relation to GRADUATION and ATTITUDE within the Appraisal framework.

The revised manuscript is now clearly structured, methodologically sound, and theoretically coherent. Your responses demonstrate strong engagement with the reviewers’ concerns, and the amendments - especially the clarification of decisions on evaluative force, the treatment of manner versus intensification, and the explanations of grammatical metaphor - significantly enhance the clarity and transparency of the study.

The contribution is valuable, and the updated version fully meets criteria (in place) for technical rigor, accuracy, and clarity. I have no remaining concerns.

Overall, I recommend "Accept."

Best regards,

Academic Editor, Dr Awni Etaywe

Reviewers' comments:

Reviewer's Responses to Questions

**Comments to the Author**

1. If the authors have adequately addressed your comments raised in a previous round of review and you feel that this manuscript is now acceptable for publication, you may indicate that here to bypass the “Comments to the Author” section, enter your conflict of interest statement in the “Confidential to Editor” section, and submit your "Accept" recommendation.

Reviewer #4: All comments have been addressed

2. Is the manuscript technically sound, and do the data support the conclusions?

Reviewer #4: Yes

3. Has the statistical analysis been performed appropriately and rigorously? 

Reviewer #4: Yes

4. Have the authors made all data underlying the findings in their manuscript fully available?

Reviewer #4: Yes

5. Is the manuscript presented in an intelligible fashion and written in standard English?

Reviewer #4: Yes

6. Review Comments to the Author

Reviewer #4: (No Response)

7. PLOS authors have the option to publish the peer review history of their article (what does this mean? ). If published, this will include your full peer review and any attached files.

**Do you want your identity to be public for this peer review?** For information about this choice, including consent withdrawal, please see our Privacy Policy .

Reviewer #4: No

---

## [Editor Report · Acceptance letter]

PONE-D-25-02479R5

PLOS ONE

Dear Dr. Chen,

I'm pleased to inform you that your manuscript has been deemed suitable for publication in PLOS ONE. Congratulations! Your manuscript is now being handed over to our production team.

Kind regards,

on behalf of

Dr. Awni Etaywe

Academic Editor

PLOS ONE